# Deep Q-Learning Based Reinforcement Learning Approach for Network Intrusion Detection

Hooman Alavizadeh [1,*] , Hootan Alavizadeh [2] and Julian Jang-Jaccard [3]

1 UNSW Institute for Cyber Security, University of New South Wales, Canberra 2612, Australia
2 Computer Engineering Department, Imam Reza International University, Mashhad 553-91735, Iran; h.alavizadeh@imamreza.ac.ir
3 Cybersecurity Laboratory, School of Information Technology and Electrical Engineering, Massey University, Auckland 0632, New Zealand; j.jang-jaccard@massey.ac.nz
* Correspondence: h.alavizadeh@adfa.edu.au

**Abstract:** The rise of the new generation of cyber threats demands more sophisticated and intelligent cyber defense solutions equipped with autonomous agents capable of learning to make decisions without the knowledge of human experts. Several reinforcement learning methods (e.g., Markov) for automated network intrusion tasks have been proposed in recent years. In this paper, we introduce a new generation of the network intrusion detection method, which combines a Q-learning based reinforcement learning with a deep feed forward neural network method for network intrusion detection. Our proposed Deep Q-Learning (DQL) model provides an ongoing auto-learning capability for a network environment that can detect different types of network intrusions using an automated trial-error approach and continuously enhance its detection capabilities. We provide the details of fine-tuning different hyperparameters involved in the DQL model for more effective self-learning. According to our extensive experimental results based on the NSL-KDD dataset, we confirm that the lower discount factor, which is set as 0.001 under 250 episodes of training, yields the best performance results. Our experimental results also show that our proposed DQL is highly effective in detecting different intrusion classes and outperforms other similar machine learning approaches.

**Keywords:** network security; deep Q networks; deep learning; reinforcement learning; network intrusion detection; NSL-KDD; artificial intelligence

## 1. Introduction

The new generation of Intrusion Detection Systems (IDSs) increasingly demands automated and intelligent network intrusion detection strategies to handle threats caused by an increasing number of advanced attackers in the cyber environment [1–3]. In particular, there have been high demands for autonomous agent-based IDS solutions that require as little human intervention as possible while being able to evolve and improve itself (e.g., by taking appropriate actions for a given environment), and to become more robust to potential threats that have not been seen before (e.g., zero-day attacks) [4].

Reinforcement Learning (RL) has become a popular approach in detecting and classifying different attacks using automated agents. The agent is able to learn different behavior of attacks launched to specific environments and formulates a defense strategy to better protect the environment in the future. An RL approach can improve its capability for protecting the environment by rewarding or penalizing its action after receiving feedback from the environment (e.g., in a trial-and-error interaction to identify what works better with a specific environment). An RL agent is capable of enhancing its capabilities over time. Due to its powerful conception, several RL-based intrusion detection techniques have been proposed in recent years to provide autonomous cyber defense solutions in various contexts and for different application scenarios such as IoT, Wireless Networks [5,6], or Cloud [7–9]. The RL agent is able to implement the self-learning capabilities during the

learning process based on its observation without any supervision requirement, which typically involves the expert knowledge from human [10]. Many network intrusion detection techniques have been proposed based on this RL concept [11]. However, most of the existing approaches suffer from the uncertainty in detecting legitimate network traffic with appropriate accuracy and further lacks the capability to deal with a large dataset. This is because an RL agent typically needs to deal with very large learning states and would encounter the state explosion problem. In recent years, deep reinforcement learning (DRL) techniques have been proposed that are capable of learning in an environment with an unmanageable huge number of states to address the main shortcoming of existing RL techniques. DRL techniques such as deep Q-learning have shown to be a promising method to handle the state explosion problem by leveraging deep neural networks during the learning process [12].

Many DRL-based IDS for network intrusion detection techniques have been proposed in the existing literature, leveraging different types of intrusion datasets to train and evaluate their models [13,14]. However, most of these existing proposals only focus on enhancing their detection ability and performance compared to other similar approaches. The majority of these existing works do not offer comprehensive studies as to how best to develop and implement a DRL-based IDS approach for a network environment without providing the precise details, such as how the DQL agent can be formulated based on an RL theory or how to fine-tune hyperparameters for more effective self-learning and interact with the underlying network environment. In this paper, we address these shortcomings by introducing the details of design, development, and implementation strategy for the next generation of the DQL approach for network intrusion detection.

The main contributions of our work are summarized as follows:

- We introduce a new generation of network intrusion detection methods that combine a Q-learning based reinforcement learning with a deep feed forward neural network method for network intrusion detection. Our proposed model is equipped with the ongoing auto-learning capability for a network environment it interacts with and can detect different types of network intrusions. Its self-learning capabilities allow our model to continuously enhance its detection capabilities.
- We provide intrinsic details of the best approaches involved in fine-tuning different hyperparameters of deep learning-based reinforcement learning methods (e.g., learning rates, discount factor) for more effective self-learning and interacting with the underlying network environment for more optimized network intrusion detection tasks.
- Our experimental results, based on the NSL-KDD dataset, demonstrate that our proposed DQL is highly effective in detecting different intrusion classes and outperforms other similar machine learning approaches, achieving more than 90% accuracy in the classification tasks involved in different network intrusion classes.

The rest of the paper is organized as follows. Section 2 presents the related work. Section 3 discusses the essential concepts and background associated with reinforcement learning and deep neural network. Section 4 represents the NSL-KDD dataset used for this paper. The proposed DQL-based anomaly detection approach is given in Section 5. The evaluation, experimental results, and analysis are given in Section 6. Finally, we conclude the paper in Section 7.

## 2. Related Work

Q-learning, considered to be a model-free method, has been hailed to be a promising approach, especially when utilized in challenging decision processes. This method is appropriate if the other techniques such as traditional optimization methods and supervised learning approaches are not applicable [15]. The advantages of Q-learning are its effective results, learning capabilities, and the potential combination with other models.

The application of machine learning such as Deep Reinforcement Learning (DRL) [16–18], supervised and unsupervised learning, in cybersecurity, has been investigated in various studies [14,15,19,20]. In [19], the authors studied a comprehensive review of DRL for

cybersecurity. They studied the papers based on the live, real, and simulated environment. In [20], the authors showed the applications of DRL models in cybersecurity. They mainly focused on the adversarial reinforcement learning methods. They also presented the recent studies on the applications of multi-agent adversarial RL models for IDS systems. In [14], the authors studied several DRL algorithms such as Double Deep Q-Network (DDQN), Deep Q-Network (DQN), Policy Gradient (PG), and Actor-Critic (AC) to intrusion detection using NSL-KDD [21] and AWID [22] datasets. Those datasets were used for training purposes and for classifying intrusion events using the supervised machine learning algorithms. They showed the advantages of using DRL in comparison with the other machine learning approaches, which are the application of DRL on modern data networks that need rapid attention and response. They showed that DDQN outperforms the other approaches in terms of performance and learning.

In [23,24], the authors proposed a deep reinforcement learning technique based on stateful Markov Decision Process (MDP), Q-learning. They evaluated the performance of their method based on different factors such as learning episodes, execution time, and cumulative reward and compared the effectiveness of standard planning-based with a deep reinforcement learning based approach.

In [25], the authors proposed a reinforcement learning agent installed on routers to learn from traffic passing through the network and avoid traffic to the victim server. Moreover, in [26], the authors proposed a machine learning method to detect multi-step attacks using hidden Markov models to predict the next step of the attacker. In [27], the authors proposed a decision-theoretic framework named ADRS based on the cost-sensitive and self-optimizing operation to analyze the behavior of anomaly detection and response systems in autonomic networks.

In [28], the authors combined the multi-objective decision problem with the evolutionary algorithms to make an efficient intrusion response system. They considered an intrusion response system as a multi-attribute decision making problem that takes account of several aspects before responding to the threats such as cost of implementation, resource restriction, and effectiveness of the time, and modification costs. This multi-objective problem tried to find an appropriate response that was able to reduce the values of these functions.

Most of the existing works focused only on the enhancement of their methods to provide better performance and on the evaluation of the performance of the proposed techniques through comparison with other similar machine learning (ML)-based approaches.

## 3. Background

### 3.1. Reinforcement Learning

Modeling a system as a Markov Decision Process (MDP) so that an agent can interact with the environment based on different discrete time steps is an important aspect in designing many decision-making related problems. MDP can be shown as a 5-tuple: M = $(S, \mathcal{A}, T, R, \gamma)$ where $S$ denotes a set of possible states and $\mathcal{A}$ indicates a set of possible actions that the agent can perform on the environment, and $T$ denotes the transition function from a state to another state. $T$ defines the (stationary) probability distribution on $S$ to transit and reach a new state $s'$. The value of $R$ denotes the reward function, and $\gamma = [0, 1)$ indicates the discount factor.

A policy $\pi$ can be defined to determine the conditional probability distribution of selecting different actions depending on each state $s$. The distribution of the reward sequence can be determined once a stationary policy has opted. Then, policy $\pi$ can be evaluated by an action-value function which can be defined under $\pi$ as the expected cumulative discounted reward based on taking action from state s and following $\pi$ policy. By solving the MDP, the optimal policy $\pi^*$ can be found that maximizes the expected cumulative discounted reward based on all states. The corresponding optimal action values satisfy $Q^*(s, a) = \max_{\pi} Q^{\pi}(s, a)$, and the uniqueness and existence of the fixed-point solution of Bellman optimality equations can be obtained by Banach's fixed-point theorem.

$$Q^*(s,a) = R(s,a) + \gamma \int_{s'} T(s'|s,a) \max_{a'} Q^*(s',a')$$

The essential cyclic process in the RL is agent-environment interaction. The RL agent should interact with the environment to explore and learn from different transition and reward functions obtained from the action taken. This process makes the RL agent able to find out the optimal policy, see Figure 1. During the interaction with the environment at time $t$, the RL agent observes the information about the current state $s$, and then chooses an action $a$ based on a policy. Then, it receives a reward $r$ from the environment based on the action taken and moves to a new state $s'$. The RL agent improves itself by experiences gained based on this cyclic agent–environment interaction. The learning process could be based on either (i) approximating the transition probabilities and reward functions to learn the MDP model and then finding an optimal policy using planning in the MDP (i.e., known as the model-based approach), or (ii) trying to learn the optimal value functions directly without learning the model and deriving the optimal policy (e.g., model-free approach).

Q-learning can be considered as a model-free approach that updates the Q-values estimation based on the experience samples on each time step as the following equation.

$$Q(s,a) \leftarrow Q(s,a) + \alpha(r + \gamma \max_{a'} Q^*(s',a') - Q(s,a))$$

where $\alpha$ is the learning rate, and $Q(s,a)$ is simply the current estimation.

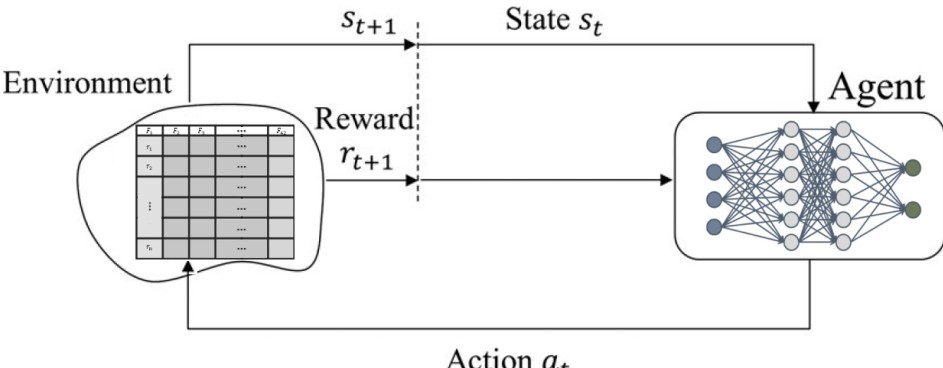

**Figure 1.** DQN model based on agent–environment interaction.

### 3.2. Feed Forward Neural Network

We utilized a feed forward neural network constructed based on a fully connected neural network as the main module of the DRL model for approximating the Q-values and training the model based on the NSL-KDD Dataset. The intrusion datasets will be fed into a pre-processing module first for cleansing and preparing the dataset and also extracting the related features [29].

The fully connected neural network includes different fully connected layers that link every single neuron in the layer to neurons of the previous layer. The neural network output $f(x)$ or $y$ is represented in Equation (1) [30].

$$y = f(x) = F_{|f|}(F_{|f|} - 1(\dots F_2(F_1(x)))) \tag{1}$$

where $x$ is the input, $F_i$ is a transformation function, and $|f|$ denotes the total number of computational layers, which can be the hidden layers and the output layer in the neural network. The outputs from the preceding layers are transferred using each perceptron by applying a non-linear activation function. Finally, the $i^{\text{th}}$ perceptron in the $t^{\text{th}}$ layer can be represented as:

$$o_i^t = \nu\left(w_{i*}^t \cdot o^{t-1} + b_i^t\right) \tag{2}$$

where $o^{t-1}$ is the output of the preceding layer, $w_{i*}^t$ is the weight vector of the perceptron, $b_i^t$ is its bias and $\nu$ is the non-linear activation function.

Activation functions play an essential role in the training process of a neural network. Activation functions manage the computations to be more effective and reasonable as there is the complexity between the input units and the response variable in a neural network. The main role of the activation function is to convert an input unit of a neural network to an output unit. Different activation functions can be used in a neural network such as Sigmoid, Tanh, and ReLU. However, the ReLU activation function is known as a more effective one compared with the other activation functions in many detection problems, with lower run-time and demands for less expensive computation costs; see Equation (3), where $z$ is the input.

$$ReLU(z) = \begin{cases} 0 & \text{if } z < 0 \\ 1 & \text{if } z \geq 0 \end{cases} \tag{3}$$

## 4. Dataset

NSL-KDD dataset is a labeled network intrusion detection dataset that so far has been used in many tasks to evaluate different deep learning-based algorithms for devising different strategies for IDS [31,32]. NSL-KDD dataset contains 41 features labeled as a normal or specific attack type (i.e., class). We utilized the one-hot-encoding method for the dataset preprocessing to change the categorical features to the corresponding numeral values as deep learning models can only work with numerical or floating values. We normalized the train and test datasets to the values between 0 and 1 using a mix-max normalization strategy. The 41 features presented in the NSL-KDD dataset can be grouped into four features, such as basic, content-based, time-based, and host-based traffic features. The value of these features is mainly based on continuous, discrete, and symbolic values. The NSL-KDD dataset contains five attack classes such as Normal Denial-of-Service (DoS), Probe, Root to Local (R2L), and Unauthorized to Root (U2R). These attack classes can be identified using the features corresponding to each NSL-KDD data. Table 1 defines the attack classes for NSL-KDD that we consider in our study.

**Table 1.** NSL-KDD data-record classes.

| Categories | Notation | Definitions | Samples # |
|---|---|---|---|
| Normal | N | Normal activities based on the features | 148,517 |
| DoS | D | Attacker tries to avoid users of a service Denial of Service attack | 53,385 |
| Probe | P | Attacker tries to scan the target network to collect information such as vulnerabilities | 14,077 |
| U2R | U | Attackers with local access to victim's machine tries to get user privileges | 119 |
| R2L | P | Attacker without a local account tries to send packets to the target host to get access | 3882 |

## 5. Anomaly Detection Using Deep Q Learning

### 5.1. Deep Q-Networks

One of the most effective types of RL is Q-learning, in which a function approximator such as either a neural network or a deep neural network is used in RL as a Q-function to estimate the value of the function. The Q-function integrated with a deep neural network can be called Deep Q-Learning (DQL). The Q-learning agent in the DQL can be represented as $Q(s, a; \theta)$. The Q-function consists of some parameters such as state $s$ of the model, action $a$, and reward $r$ value. The DQL agent can select an action $a$ and correspondingly receives a reward for that specific action. The neural network weights related to each layer in the Q-network at time $t$ are denoted by the $\theta$ parameter. Moreover, $s_{i+1}$ or $s'$ represents the next state for the DQL model. The DQL agent moves to the next state based on the previous state $s$ and the action $a$ performed in the previous state $s$. A deep neural network is used as

the deep Q-network to estimate and predict the target Q-values. Then, the loss function for each learning activity can be determined by Q-values obtained on the current and previous states. In some cases, only one neural network is used for estimating the Q-value. In this case, a feedback loop is constructed to estimate the target Q-value so that the target weights of the deep neural network are periodically updated.

### 5.2. Deep R-Learning Concepts

Here we define the important concepts related to DQL based on the environment where the NSL-KDD dataset is used for network intrusion detection tasks.

#### 5.2.1. Environment

The environment for this study is the one where the pre-processed and normalized NSL-KDD dataset is used, where the columns (features) of the NSL-KDD dataset denote the states of the DQN. There are 42 features in NSK-KDD and we utilize the first 41 features as states. Feature 42 is the label that will be used for computing the award vectors based on model prediction. Note that in this DQN model, the agent only obtains actions to compute the rewards vector, and there is no real action performed to the environment.

#### 5.2.2. Agent

DQL agent is utilized in the DQL model based on the network structure so that there is at least one agent for the context of a network. The agent interacts with the environment and applies rewards based on the current state and the selected action. A DQL agent could be defined as a value-based RL agent that is able to train the model to estimate the future rewards values. A DQN can be trained by an agent interacting with the environment based on the observation and possible action spaces. In the DQL training process, the agent needs to explore the action space by applying a policy such as epsilon-greedy exploration. The exploration helps the agent to select either a random action with a probability of $\epsilon$ or an action greedily based on the value function with the greatest value with probability $1 - \epsilon$.

#### 5.2.3. States

States in DQL describe the input by the environment to an agent for taking action. In the environment where the NSL-KDD dataset is used, the dataset features (as in Table 2) are used for state parameters for DQN. We use those 41 features as the inputs of DQN such that $s_i = F_i$ for training and prediction using DQN.

**Table 2.** List of features on NSL-KDD dataset.

| F# | Feature Name | F# | Feature Name | F# | Feature Name |
|----|--------------|----|--------------|----|--------------|
| F1 | Duration | F15 | Su attempted | F29 | Same srv rate |
| F2 | Protocol_type | F16 | Num root | F30 | Diff srv rate |
| F3 | Service | F17 | Num file creation | F31 | Srv diff host rate |
| F4 | Flag | F18 | Num shells | F32 | Dst host count |
| F5 | Src bytes | F19 | Num access files | F33 | Dst host srv count |
| F6 | Dst bytes | F20 | Num outbound cmds | F34 | Dst host same srv rate |
| F7 | Land | F21 | Is host login | F35 | Dst host diff srv rate |
| F8 | Wrong fragment | F22 | Is guest login | F36 | Dst host same srv port rate |
| F9 | Urgent | F23 | Count | F37 | Dst host srv fiff host rate |
| F10 | Hot | F24 | Srv count | F38 | Dst host serror rate |
| F11 | Num_failed_logins | F25 | Serror rate | F39 | Dst host srv serror rate |
| F12 | Logged_in | F26 | Srv serror rate | F40 | Dst host rerror rate |
| F13 | Num compromised | F27 | Rerror rate | F41 | Dst host srv rerror rate |
| F14 | Root shell | F28 | Srv rerror rate | F42 | Class label |

### 5.2.4. Actions

An action is considered as the decision chosen by the agent after processing the environment during a given time window such as after finishing the process of a mini-batch. The DQN agent generates a list of actions as an action vector based on the given input of the neural network and input features. The final Q-values are used to judge whether an attack was captured successfully. It feeds the state vector with the size of the mini-batch to the current DQN. Then, the agent compares the output of the current DQN based on threshold rates as Q-values and determined the Q-threshold value for classifying the attack classes.

### 5.2.5. Rewards

In DQL, the feedback from the environment for a corresponding action done by an agent is called a reward. A reward vector can be defined based on the output values of DQN and the size of the mini-batch. The DQL agent can consider a positive reward when the classification result of DQN matches the actual result based on the labels in the NSL-KDD. Otherwise, it may get a negative reward. The reward value can be considered depending on the probability of prediction by the classifier. This value can be adjusted based on the Q-values obtained to enhance the classifier's performance.

### 5.3. Deep Q-Learning Process

The standard Q-learning and DQN can be differentiated based on the method of estimating the Q-value of each state-action pair and the way in which this value can be approximated using generalized state-action pair by the function $Q(s,q)$. The process of DQL is based on the Pseudocode represented in Algorithm 1 that describes how the DQN agent deals with the environment using the NSL-KDD. In the first step, the parameters of the algorithm and models are initialized based on Table 3. The values of the features of NSL-KDD (F1–F41 as in Table 1) indicate the state's variables ($s$) of the DQN. Note that the batch size ($bs$) for the DQN process is set as 500. This means that for each state the amount of 500 records of NSL-KDD are fetched from memory and fed into one state ($S$), see the batch table represented in Figure 2. However, there are 41 features as the state variables, each of which can have various values. Thus, as the number of state-value pairs becomes comparatively large, it is not possible to keep them in a Q-table (or look-up table). Thus, the DQN agent leverages a DNN as the function approximator to compute Q values based on the states and actions.

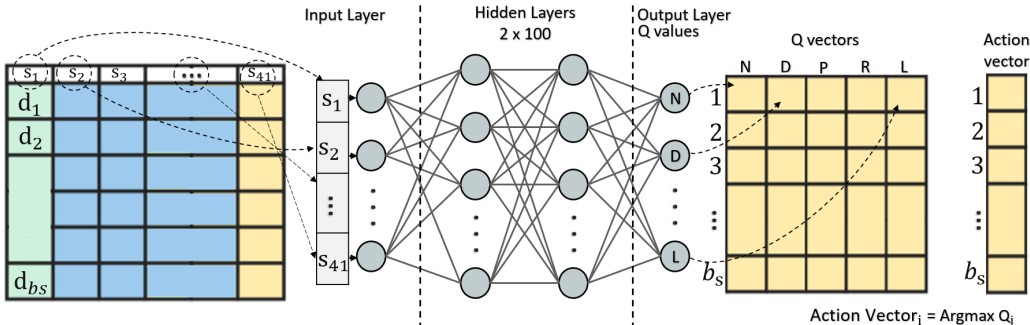

**Figure 2.** DQN model prediction using states and deep neural network, the outputs are Q-values, and actions are computed based on *argmax* $Q_i$ for the current state.

**Table 3.** DQL agent and Neural Network parameters.

| Parameters | Description | Values |
|---|---|---|
| num-episode | Number of episodes to train DQN | 200 |
| num-iteration | Number of iteration to improve Q-values in DQN | 100 |
| hidden_layers | Number of hidden layers: Setting weights, producing outputs, based on activation function | 2 |
| num_units | number of hidden unit to improve the quality of prediction and training | $2 \times 100$ |
| Initial weight value | Normal Initialization | Normal |
| Activation function | Non-linear activation function | ReLU |
| Epsilon $\epsilon$ | Degree of randomness for performing actions | 0.9 |
| Decoy rate | Reducing the randomness probability for each iteration | 0.99 |
| Gamma $\gamma$ | Discount factor for target prediction | 0.001 |
| Batch-size (bs) | A batch of records NSL-KDD dataset fetched for processing | 500 |

---

**Algorithm 1:** Deep Q-learning agent training based on NSL-KDD environment.

---

**Data:** NSL-KDD dataset                      `/* Environment */`
**Data:** DQL parameters                    `/* Agent parameters */`
       `/* Result:  Dictionaries of VMs and Reachability, VMs and Vulnerabilities */`
**begin**
                         `/* Pre-processing and Parameters Initialization */`
    Normalize(NSL-KDD)
    Initialize parameters←as Table 3
    bs←500
    State←fetch(NSL-KDD,bs)
    Create_model(States, hidden_layers, ReLU, output_layers)
    `/* DQL agent learning episodes and iterations */`
    **foreach** *epoch* ∈ *num-episode* **do**
       Reset(states)
       Create(q_val_List[size=bs,Action_size])
       **foreach** *T* ∈ *num_iteration* **do**
          Initialize parameters
          `/* With probability of ` $\epsilon$ `:  */`
          $AV_i \leftarrow$ Create_random(Action_space) $\forall i \in bs$
          $\epsilon \leftarrow \epsilon * decoy\_rate$
          `/* With probability of ` $1 - \epsilon$ `:  */`
          $QV_i \leftarrow$ model.predict(current-state)
          $AV_i =$ Argmax($QV_i$) $\forall i \in bs$
          `/* Compute rewards */`
          $RV_i \leftarrow$ Compute_reward(AV,labels) $\forall i \in bs$
          `/* Agent's learning improvement */`
          $Q' \leftarrow$ model.predict(state')
          $QT_i \leftarrow RV_i + \gamma * Q[state', action']$
          Model.train(state,$QT_i$)
          *Compute_loss*($QV_i$,$QT_i$)
          *State←State'*
       **end**
    **end**
**end**

---

Figure 3 presents the overall steps of DQL using an agent. First, the normalized NSL-KDD dataset is fed into the environment and the DQL agent initializes a vector for Q values, state's variables, actions according to batch size, the DNN parameters, and weights are initialized. Then, the learning iterations train the DQN based on the epsilon-greedy approach. The outer iteration represents different episodes of the learning process and after each iteration, the value of states is initialized again. Note that the parameters of the trained DNN are preserved and are not initialized for each episode iteration. The state $S_n$ at each discrete state is given by the training sample (Batch$_n$), see Figure 4. At the end of each

episode, a complete sequence of states, rewards, and actions are obtained in the terminal state. During the start of the training, the agents receive the first batch (500 records from the environment), and this is the starting state $S_1$ of the environment.

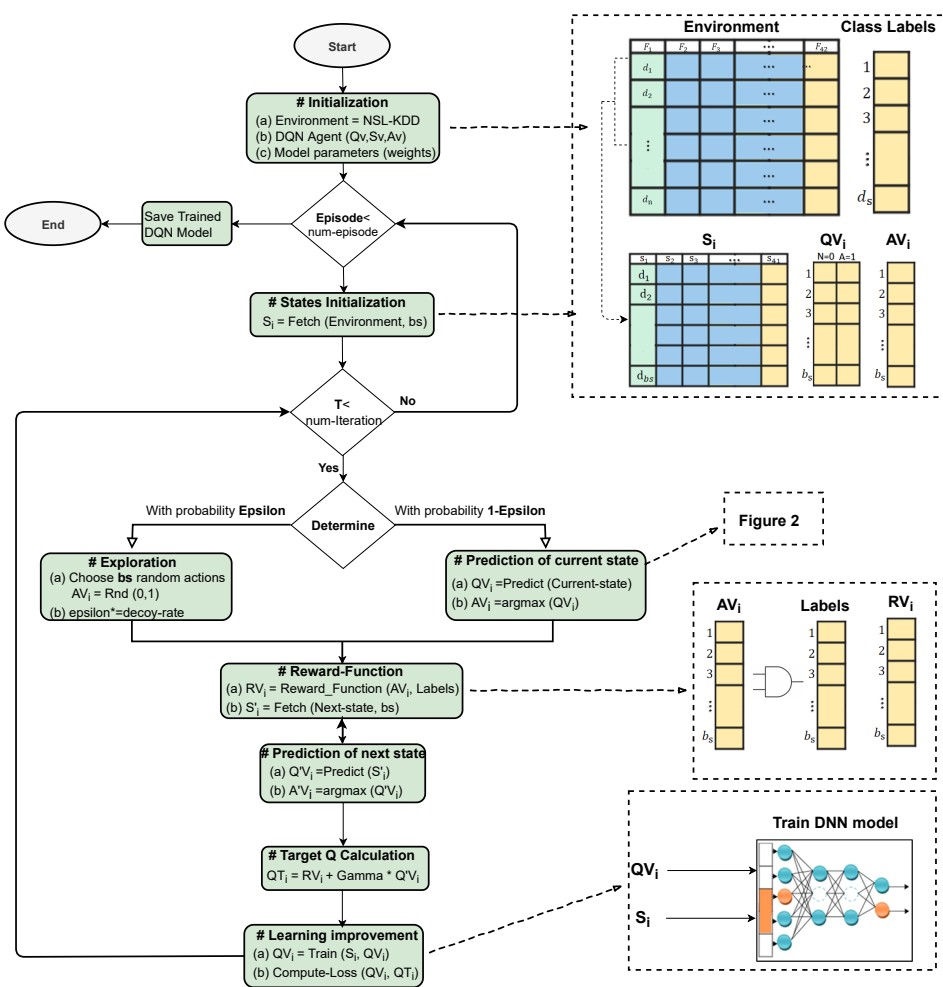

**Figure 3.** DQL agent training phase flowchart.

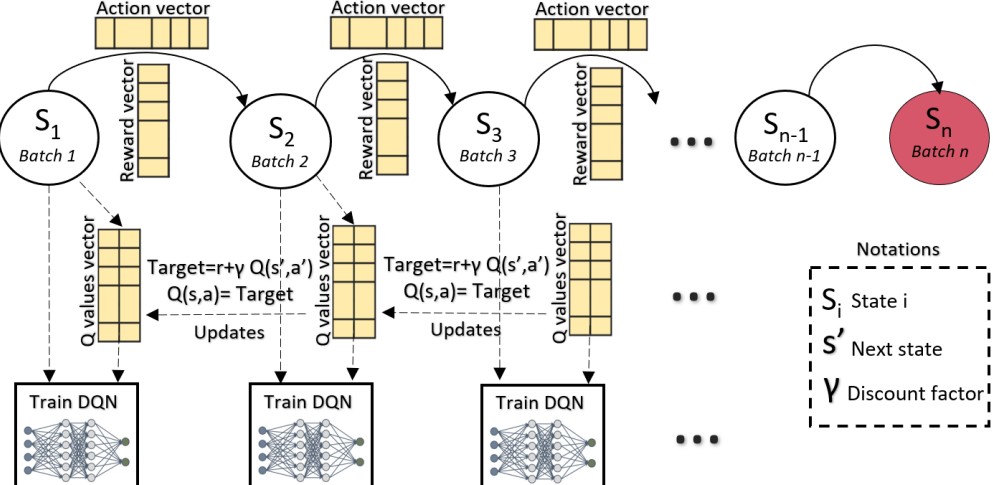

**Figure 4.** State transition Markov diagram for DQN agent training process based on current and next states prediction and training.

## 6. Evaluation of the DQL Model

In the inner iteration, the DQN agent performs exploration, action selection, and model training based on DQL. Note that each iteration adjusts the Q-function approximator which uses a DNN. In the standard Q-learning, a table of values is kept and the agent separately updates each state-value pair. However, DQN utilizes a deep-learning approach to estimate the Q-function. We leverage a deep neural network as a function approximator for the Q-function. We use a deep neural network consisting of 4-layers as represented in Figure 2, with ReLU activation for all layers, including the last one to ensure a positive Q-value. Layer one is the input layer, which includes 41 neurons and is fed with the state variables on each iteration. There are 2 hidden layers with the size of 100 each for training purposes, and 1 output layer with the size of 5, which keeps the output layer corresponding to the related Q values for each attack class. Note that after each training iteration based on the states and batch size, the Q values predicted in the output later will be fed into Q vectors (denoted as QV) as illustrated in Figure 2.

In the DQN learning procedure, there should be an appropriate trade-off between exploitation and exploration. At the first rounds of learning, the exploration rate should be set as a high probability with the value of approximately 1, and gradually decreases using a decoy rate, see Figure 5. The exploration is performed based on the epsilon-greedy approach. An epsilon-greedy policy is implemented as a training strategy based on reinforcement learning definition which helps the agent to explore all possible actions and find the optimal policy as the number of explorations increases. The action is chosen by applying the epsilon-greedy approach, which selects a random action with a probability of $\epsilon$ or predicts the action with a probability of $(1 - \epsilon)$.

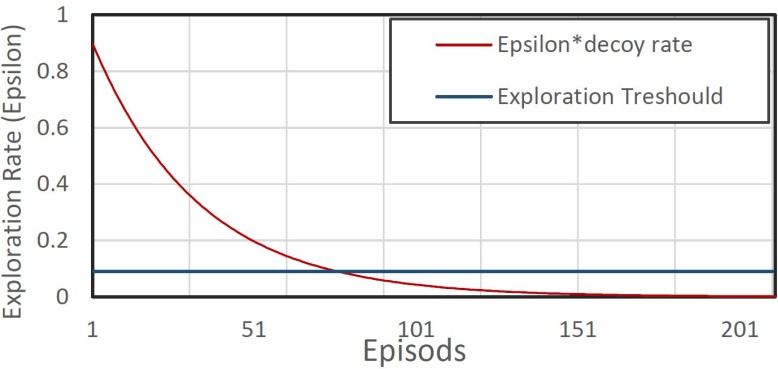

**Figure 5.** Exploration strategy using Epsilon-greedy approach.

As the batch size for each state is equivalent to *bs* (defined in Table 3), *bs* numbers of random actions will be fed into the action vector (AV) as $AV_i = Rnd(0, 5)$, $\forall i \in bs$, where 0–5 denotes Norman (N), Probe (P), DoS (D), U2R (R), and R2L (R), respectively. In the first set of iterations, the probability of choosing random actions is high, but as time passes this probability gets lower by the epsilon-greedy approach, as in Figure 5. With the probability of $1 - \epsilon$, the DQL agent predicts the actions using the current state (including the first batch with the size of *bs* and state variables). Note that, the amount of *bs* records of the environment denotes the current state. The features (i.e., variable states) of the current state are fed into the input layer of DNN architecture and the Q-values are predicted based on the DNN parameters and weights in the output layer. The action having the highest Q values is selected for each record *i* in the current state as $AV_i = argmax(QV_i)$, $\forall i \in bs$, which can be either normal or malicious based on the four attack types.

In the next step, the action vector (AV) filled by either random actions (i.e., with a higher chance in the first iterations) or predicted actions using DQN (i.e., with higher chance after a while) is fed into the reward function for computing the rewards based on comparing the $AV_i$ with the labels in the dataset for corresponding data-record, see the reward function represented in Figure 3. Then, the DQL agent needs to compute the Q vectors and action vectors of the next state denoted as $Q'V_i$ and $A'V_i$ for all $i \in bs$ to

complete the training process and DQL principles as illustrated in Figure 4. Then, the target Q (denoted as QT) is computed based on the rewards, discount factor for future rewards, and predicted Q vectors as Equation (4).

$$QT_i = RV_i + \gamma \cdot Q'V_i \tag{4}$$

The results of $QT_i$ are further fed into the DQN for the training process and computing the loss function as represented in the learning improvement phase in Figure 3. The training of the neural network is performed with a Mean Square Error (MSE) loss between the Q-value estimated by the neural network for the current state and a target Q value obtained by summing the current reward and the next state's Q-value multiplied by the value of discount factor ($\lambda$). The computation of the loss function for the evaluation of the DQN performance is critical. We compute the loss value after each iteration episode for the DQN network based on the current states and target network. The total loss can be denoted as Equation (5).

$$Loss = \frac{1}{n} \sum_n \left( \underbrace{Q(s,a)}_{\text{Prediction}} - \underbrace{r + \gamma Q(s',a')}_{\text{Target}} \right)^2 \tag{5}$$

Once the training of the model is completed, the trained NN is used for prediction. For each state, the Q-function provides the associated Q-value for each of the possible actions for that specific state. The predicted action is determined based on the maximum Q-value. The model is trained for a number of iterations and episodes, which are enough for covering the complete dataset.

### 6.1. Experiment Setup and Parameters

We implemented our proposed model in Python using Tensorflow framework version 1.13.2. In our study, we analyzed the performance of our DQL model using NSL-KDD datasets. The training portion of the NSL-KDD dataset includes various samples for the network features and corresponding labels for intrusion with different possible values such as binary or multiclass anomaly. In this paper, we considered the network features as states and the label values as the actions to adapt these elements to DQN concepts.

The parameters and values associated with the DQL model are shown in Table 3. For the fully connected architecture, we used a total of two hidden layers with 'relu' activation function apart from input and output layers. During the training, various major parameters should be determined and examined in order to find the best values that are appropriate and ideal for the model. At the beginning step for training, the exploration rate $\epsilon$, is set to 0.9 for the agent to perform exploration based on some degree of randomness with a decoy rate of 0.99. The initial values for other values such as batch-size and discount factor are illustrated in Table 3. However, we also evaluate the performance of the DQL model based on different values. We examined the behavior of the proposed DQL agent by varying the discount factor values. This essentially determines how the DQL agent can improve the performance of learning based on future awards.

Figure 6 demonstrates the loss and reward values obtained during the DQL training process based on different values of the discount factor. Figure 6a,b show the reward and loss values by setting the $\lambda$ as 0.001 and 0.01, respectively. As it shows, the loss value is lower in $\lambda = 0.001$. However, we increased the learning rate significantly and evaluated the loss and reward values in Figures Figure 6c,d. The results indicate that a higher value for the discount factor leads to a higher loss value. As it also shows, the loss value in the worst-case reaches 1.7 based on $\lambda = 0.001$, while it reaches 4 in the higher discount factor value $\lambda = 0.9$. However, as demonstrated in Figure 6, we can observe that reward values have a sharp increasing trend for all discount factor values.

Based on the results obtained during the DQN agent learning process, we discovered that the lower discount factor yields a lower loss value that leads to better results in terms of learning the model, especially when the episode numbers are smaller.

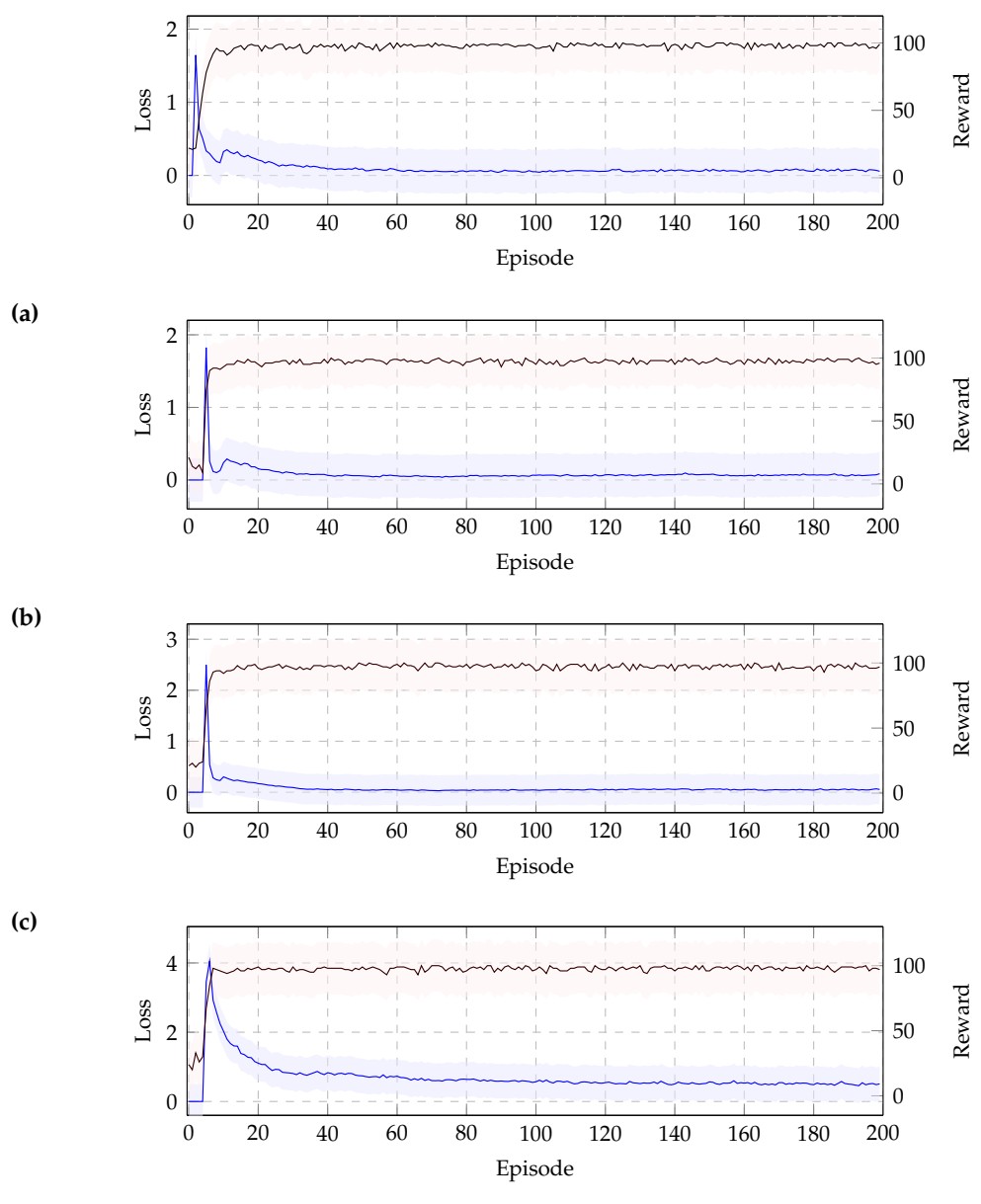

**Figure 6.** Comparing the loss and reward values of DQN learning process based on different discount factor values: (**a**) $\gamma = 0.001$, (**b**) $\gamma = 0.01$, (**c**) $\gamma = 0.1$, and (**d**) $\gamma = 0.9$.

### 6.2. Performance Metrics

We use different measurements to evaluate the performance of our proposed DQL model used for network intrusion detection, such as Accuracy, Precision, Recall, and F1 score. However, the performance of the model cannot rely only on the accuracy values since it evaluates the percentages of the samples that are correctly classified. It ignores the samples incorrectly classified. To perform better evaluation, we analyzed the results based on the other performance metrics, as follows.

#### 6.2.1. Accuracy

Accuracy is one of the most common metrics to evaluate and judge a model. It measures the total number of correct predictions made out of all the predictions made by the model. It can be obtained based on True Positive (TP) value, True Negative (TN)

rate, False Positive (FP) rate, and False Negative (FN) value. Equation (6) shows the Accuracy metric.

$$\text{Accuracy} = \frac{\text{TP} + \text{TN}}{\text{TP} + \text{FP} + \text{TN} + \text{FN}} \tag{6}$$

6.2.2. Precision

Precision evaluates can be obtained based on the percentage of positive instances against the total predicted positive instances. In this case, the denominator is the sum of TP and FP, denoting the model prediction performed as positive from the whole dataset. Indeed, it indicates that 'how much the model is right when it says it is right', see Equation (7).

$$\text{Precision} = \frac{\text{TP}}{\text{TP} + \text{FP}} \tag{7}$$

6.2.3. Recall

Recall (Sensitivity) shows the percentage of positive instances against the total actual positive instances. The denominator is the sum of TP and FN values, which is the actual number of positive instances presented in the dataset. It indicates 'how many right ones the model missed when it showed the right ones'. See Equation (8).

$$\text{Recall} = \frac{\text{TP}}{\text{TP} + \text{FN}} \tag{8}$$

6.2.4. F1 Score

The harmonic mean of precision and recall values is considered as the F1 score. It considers the contribution of both values. Thus, a higher F1 score indicates better results. Based on the numerator of Equation (9), if either precision or recall value goes low, the final value of the F1 score also decreases significantly. We can conclude a model as a good one based on the higher value of the F1 score. Equation (9) shows how the F1 score is computed based on both precision and recall values.

$$\text{F1 score} = \frac{2 \times \text{Precision} \times \text{Recall}}{\text{Precision} + \text{Recall}} \tag{9}$$

*6.3. Performance Evaluation*

We evaluated the performance of the DQN on the testing phase based on the parameters set in Table 3 and training based on 200 episodes.

The confusion matrix for the DQL model based on two different discount factors of 0.001 and 0.9 are shown in Figure 7. The confusion matrix represented the evaluation of our model for the test data set. The rows in the confusion matrix are associated with the predicted class and the columns indicate the true class. The confusion matrix cells on the main diagonal demonstrate the correctly classified percentages such as those having been classified as TP or TN. However, the incorrectly classified portion is located in the off-diagonal cells such as FN and FP values. The values located on the last columns (most right columns) indicate the percentages of incorrectly classified predictions corresponding to each class. Considering the confusion matrices, we observe that the true-positive rate for normal, DoS, and Probe classes for $\lambda = 0.001$ has decreased from 0.96 to 0.82, and 0.68 to 0.93, 0.89, and 0.57 for $\lambda = 0.9$, respectively. This shows that the DQL agent performs better for the smaller values of discount factor $\gamma = 0.001$ compared to larger discount factor value such as $\gamma = 0.9$. However, the results for the minority class of R2L are very low because of unbalanced distributions of the number of samples for each class (see Table 1).

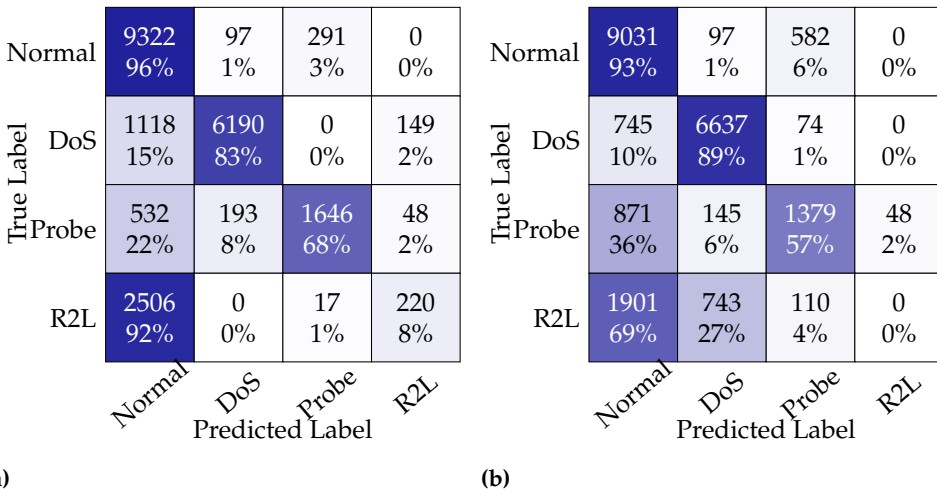

(a)                                                                  (b)

**Figure 7.** Confusion matrix based on the classification categories for our DQL model for two different discount factors: (**a**) $\gamma = 0.001$, (**b**) $\gamma = 0.9$.

Figure 8 shows the performance of the proposed DQL model for the correct estimations (TP and TN) together with FP and FN values based on the number of samples. The results are captured after performing 200 episodes for the agent's learning with the discount factor of $\lambda = 0.001$. Considering the graph, we observe higher correct estimations for Normal, DoS, and Probe classes, respectively, while these values are lower for the minority classes due to the unbalanced distribution of class samples.

We evaluated the performance or DQL model based on both accuracy and time against the different numbers of training episodes in Figure 9. We can observe that the accuracy value has an ascending trend from 100 episodes to 250 episodes. However, this value decreased to 300 episodes, while training based on 300 episodes lasts more than 20 min. It shows that the best number of episodes for the DQL agent for training is 250 episodes, which take a smaller execution time of around 17 min based on our implementation.

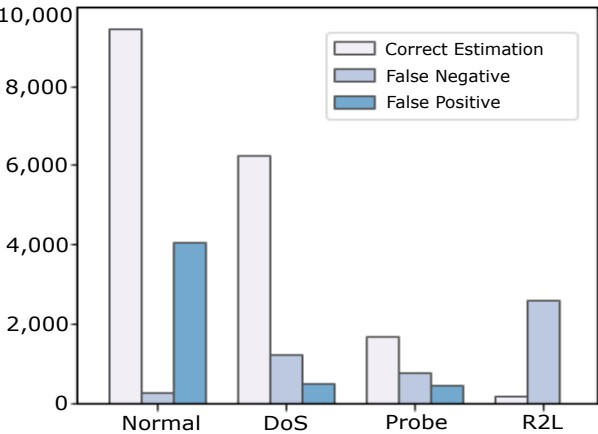

**Figure 8.** # of samples against estimated attack types.

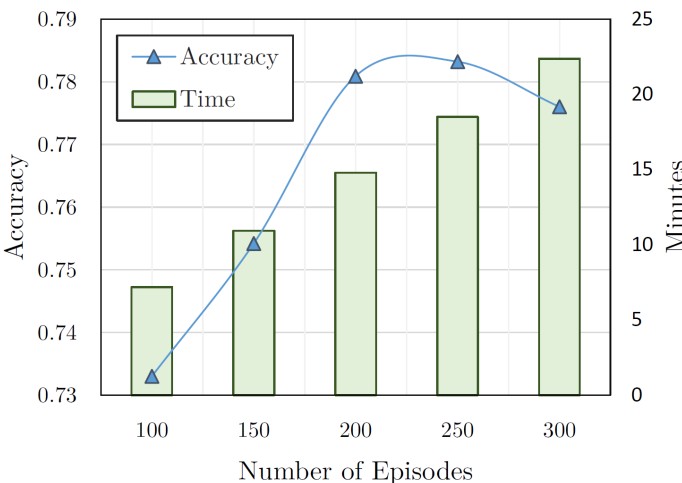

**Figure 9.** Performance of the DRL process based on different episodes.

Table 4 compares the overall performance of the DQL based on two different discount factors in the DQL training process. The results show that all performance metrics have higher values for the smaller value of discount factor $\lambda = 0.001$. Table 5 shows the performance metrics for each class separately while the DQL model is trained based on 200 episodes with the discount factor $\lambda = 0.001$.

**Table 4.** Performance evaluation of DQL based on various discount factor values.

| Metric | Discount Factors | | |
|---|---|---|---|
| | $\gamma = 0.001$ | $\gamma = 0.1$ | $\gamma = 0.9$ |
| Precision | 0.7784 | 0.6812 | 0.6731 |
| Recall | 0.7676 | 0.7466 | 0.758 |
| F1 score | 0.8141 | 0.7063 | 0.6911 |
| Accuracy | 0.7807 | 0.7473 | 0.7578 |

**Table 5.** Evaluation metrics for DQL Model based on each classes.

| Metric | Attack Categories | | | |
|---|---|---|---|---|
| | Normal | DoS | Probe | R2L |
| Accuracy | 0.8094 | 0.9247 | 0.9463 | 0.8848 |
| F1 score | 0.8084 | 0.9237 | 0.9449 | 0.8370 |
| Precision | 0.8552 | 0.9249 | 0.9441 | 0.8974 |
| Recall | 0.8093 | 0.83 | 0.9247 | 0.8848 |

*6.4. Comparison with Other Approaches*

In this section, we compare the results obtained from our proposed DRL models with various common ML-based models based on NSL-KDD datasets. We compare the results with Self-organizing Map (SOM), Support Vector Machine (SVM), Random Forest (RF), Naive Bayes (NB), Convolutional Neural Network (CNN), and some hybrid models such as BiLSTM and CNN-BiLSTM models presented in different studies [33–35].

We compare the results based on the performance metrics such as Accuracy, Recall, F1 Score, and Precision. Table 6 compares the accuracy and training time (in minutes) with other studies in the literature.

As it shows, our model has a higher accuracy compared with the other approaches, while it has a lower training time. However, the worst accuracy obtained by the SVM approach is about 68%. Both BiLSTM and CNN-BiLSTM hybrid approaches have high accuracy of 79% and 83%, respectively. However, those approaches have a higher training time than our model.

Table 7 compares our model's performance in terms of F1-score for all classifications with other studies in the literature. F1-score is known as a balance point between recall and precision scores and can be considered as the harmonic average of both recall and precision. Based on the results summarized in Table 7, the F1-score for the Normal class reaches about 81% in our study while this value is higher for CNN and CNN-BiLSTM hybrid approaches, with values of around 90%. However, it can be seen that our models perform better in terms of other attack classes such as DoS, Probe, R2L compared with other ML-based and hybrid approaches in the literature. It can be seen that the RF method has the lowest value compared to the other approaches.

**Table 6.** Comparing our model's accuracy and time with other studies in the literature.

| Approach | Reference | Adaptive-Learning | Dataset | Accuracy | Time |
|----------|-----------|-------------------|---------|----------|------|
| SOM | Ibrahim et al. [33] | ✗ | NSL-KDD | 75% | NA |
| RF | Jiang et al. [34] | ✗ | NSL-KDD | 74% | NA |
| BiLSTM | | ✗ | NSL-KDD | 79% | 115 |
| CNN-BiLSTM | | ✗ | NSL-KDD | 83% | 72 |
| Naive Bayes | Yang et al. [35] | ✗ | NSL-KDD | 76% | NA |
| SVM | | ✗ | NSL-KDD | 68% | NA |
| DQL | Our model | ✓ | NSL-KDD | 78% | 21 |

**Table 7.** Comparing our model's performance in classification with other studies in the literature

| Approach | Reference | Normal | DoS | Probe | R2L |
|----------|-----------|--------|-----|-------|-----|
| RF | Jiang et al. [34] | 0.7823 | 0.8695 | 0.7348 | 0.0412 |
| CNN | Yang et al. [35] | 0.9036 | 0.9014 | 0.6428 | 0.1169 |
| LSTM | | 0.8484 | 0.8792 | 0.6374 | 0.0994 |
| CNN-BiLSTM | | 0.9215 | 0.8958 | 0.7111 | 0.3469 |
| DQL | Our model | 0.8084 | 0.9237 | 0.9463 | 0.8848 |

## 7. Conclusions

We present a Deep Q-learning based (DQL) reinforcement learning model to detect and classify different network intrusion attack classes. The proposed DQL model takes a labeled dataset as input, then provides a deep reinforcement learning strategy based on deep Q networks.

In our proposed model, a Q-learning based reinforcement learning is combined with a deep feed-forward neural network to interact with the network environment, where network traffic is captured and analyzed to detect malicious network payloads in a self-learning fashion by DQL agents using an automated trial-error strategy without requiring human knowledge. We present the implementation of our proposed method in detail, including the basic elements of DQL such as the agent, the environment, together with the other concepts such as the quality of actions (Q-values), epsilon-greedy exploration, and rewards. To enhance the learning capabilities of our proposed method, we analyzed various (hyper) parameters of the DQL agent such as discount factor, batch size, and the number of learning episodes to find the best fine-tuning strategies to self-learn for network intrusion tasks.

Our experimental results demonstrated that the proposed DQL model can learn effectively from the environment in an autonomous manner and is capable of classifying different network intrusion attack types with high accuracy. Through the extensive experiments on parameter fine-tuning, we confirmed that the best discount factor for our proposed method should be 0.001 with 250 episodes of learning.

For future work, we plan to deploy our proposed method on a realistic cloud-based environment [36,37] to enable the DQL agent to improve its self-learning capabilities and classify the threats with high accuracy in a real-time manner. We plan to apply our proposed model in improving the self-learning capabilities in detecting Android-based malware [38] and ransomware [39] to test the generalizability and practicability of our model. We also plan to deploy our proposed model in other applications such as outlier detection of indoor air quality applications [40].

**Author Contributions:** Conceptualization, H.A. (Hooman Alavizadeh) and H.A. (Hootan Alavizadeh); methodology, H.A. (Hooman Alavizadeh); software, H.A. (Hootan Alavizadeh) and H.A. (Hooman Alavizadeh); validation, H.A. (Hooman Alavizadeh) and J.J.-J.; formal analysis, H.A. (Hooman Alavizadeh); investigation, H.A. (Hooman Alavizadeh), J.J.-J.; resources, H.A. (Hooman Alavizadeh), J.J.-J.; data curation, H.A. (Hooman Alavizadeh) and H.A. (Hootan Alavizadeh); writing—original draft preparation, H.A. (Hooman Alavizadeh) and H.A. (Hootan Alavizadeh); writing—review and editing, H.A. (Hooman Alavizadeh), J.J.-J.; visualization, H.A. (Hooman Alavizadeh) and H.A. (Hootan Alavizadeh); supervision, J.J.-J.; project administration, H.A. (Hooman Alavizadeh). All authors have read and agreed to the published version of the manuscript.

**Funding:** This work was supported by the Cyber Security Research Programme—"Artificial Intelligence for Automating Response to Threats" from the Ministry of Business, Innovation, and Employment (MBIE) of New Zealand as a part of the Catalyst Strategy Funds under Grant MAUX1912.

**Institutional Review Board Statement:** Not applicable.

**Informed Consent Statement:** Not applicable.

**Data Availability Statement:** Data is contained within the article.

**Conflicts of Interest:** The authors declare no conflict of interest.

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
