# Peer review of "Deep Q-Learning Based Reinforcement Learning Approach for Network Intrusion Detection"

_computers, doi:10.3390/computers11030041_

Round 1
Reviewer 1 Report
The article introduced a new generation of network intrusion detection methods that combines a Q-learning-based reinforcement learning with a deep feed-forward neural network method for network intrusion detection.
This approach is good and used very well in this study in comparison with other studies. Overall, the article is good to publish with Computers journal. However, a minor comment about the resolution of the figures; Figure 5, 6, and 8 needs to improve the resolution to be more clear for the readers
Author Response
We thank the reviewer for the comments. We have now changed the Figures 5 and 8 formats from jpg to eps (vector graph figure.) and also enlarge the figures' sizes to make them more readable by the readers. We have also enlarged the figure 6 size.
Reviewer 2 Report
OK, I think it is well written in general.
However, I have some remarks to the contribution structure:
- the structure seems for a scientific paper and for readers a little complicated.
1. Introduction, maybe 1.1 related work, 1.2 background
Next, I would expect 2. methods...
3. Data and experiments
4. anomaly detection using DQL
your chapter 5 is too complicated, maybe put here a schema, flowchart
Evaluation of experiments and conclusion - ok
The last 7 citations contain Jang-Jaccard, is it necessary?
Author Response
We thank the reviewer for the valuable comments, we have tried our best to address the concerns raised by the reviewer. Please find our responses to your comments below.
We acknowledge and appreciate the expert view of the reviewer regarding our paper as a scientific paper. However, as the reviewer indicated, this paper is conducted for the scientific readers in the field of AI and Cybersecurity. However, we tried to make it easier for the readers to understand the DQL section (Section 5) which is more specialized. As the reviewer suggested, we have now added pseudocode in Algorithm 1 to describe the process of DQL in detail besides the flowchart presented in Figure 4. Moreover, we have also removed the additional references that are less related to this paper.